# Amino acid sequence assignment from single molecule peptide sequencing data using a two-stage classifier

**Matthew Beauregard Smith**[1]*, **Zack Booth Simpson**[2]*, **Edward M. Marcotte**[3]*

**1** Oden Institute, The University of Texas at Austin, Austin, Texas, United States of America, **2** Erisyon Inc., Austin, Texas, United States of America, **3** Department of Molecular Biosciences, The University of Texas at Austin, Austin, Texas, United States of America

* mbsmith93@utexas.edu (MBS); zack@erisyon.com (ZBS); marcotte@utexas.edu (EMM)

**Data Availability Statement:** This paper describes capabilities of whatprot, which is an open source

## Abstract

We present a machine learning-based interpretive framework (*whatprot*) for analyzing single molecule protein sequencing data produced by fluorosequencing, a recently developed proteomics technology that determines sparse amino acid sequences for many individual peptide molecules in a highly parallelized fashion. Whatprot uses Hidden Markov Models (HMMs) to represent the states of each peptide undergoing the various chemical processes during fluorosequencing, and applies these in a Bayesian classifier, in combination with prefiltering by a k-Nearest Neighbors (kNN) classifier trained on large volumes of simulated fluorosequencing data. We have found that by combining the HMM based Bayesian classifier with the kNN pre-filter, we are able to retain the benefits of both, achieving both tractable runtimes and acceptable precision and recall for identifying peptides and their parent proteins from complex mixtures, outperforming the capabilities of either classifier on its own. Whatprot's hybrid kNN-HMM approach enables the efficient interpretation of fluorosequencing data using a full proteome reference database and should now also enable improved sequencing error rate estimates.

## Author summary

Scientists often wish to know which proteins, and at what quantities, are present in a sample. The field of proteomics offers a number of technologies that aid in this, such as tandem mass spectrometry and immunoassays, that provide different tradeoffs between sensitivity, throughput, and generality. One new technology, known as fluorosequencing, detects and provides partial sequences for individual peptide or protein molecules from a sample in a highly parallelized fashion. However, as only partial sequences are measured, the resulting sequencing reads must be matched to a reference database of possible proteins, such as might be obtained from the human genome. We describe a suitable computer algorithm for performing this matching of fluorosequencing reads to a reference database while accounting for the most prevalent types of sequencing errors. We detail its performance and implementation, and describe a number of uncommon algorithmic

software tool available at https://github.com/marcottelab/whatprot.

**Funding:** M.B.S. acknowledges support from a Computational Sciences, Engineering, and Mathematics graduate program fellowship. Z.B.S. acknowledges support from Erisyon, Inc. E.M.M. acknowledges support from Erisyon, Inc., the National Institute of General Medical Sciences (R35GM122480), the National Institute of Child Health and Human Development (HD085901), and the Welch Foundation (F-1515). The funders had no role in study design, data collection and analysis, decision to publish, or preparation of the manuscript.

**Competing interests:** I have read the journal's policy and the authors of this manuscript have the following competing interests: E.M.M and Z.B.S. are co-founders and shareholders of Erisyon, Inc., and are co-inventors on granted patents or pending patent applications related to single-molecule protein sequencing. Z.B.S. is an employee of Erisyon, Inc. E.M.M. serves on the scientific advisory board.

improvements and approximations which allow this approach to scale to classification against the whole human proteome. The resulting software, known as whatprot, allows researchers to interpret fluorosequencing reads and better apply this emergent single molecule protein sequencing technology.

This is a *PLOS Computational Biology* Methods paper.

## Introduction

Proteins are key components of living organisms, but their heterogenous chemical natures often complicate their biochemical analyses, and consequently, the state of protein identification and quantification methods (e. g., mass spectrometry, antibodies, affinity assays) has generally tended to lag the remarkable progress exhibited by DNA and RNA sequencing technologies. However, improvements to protein analyses could potentially directly inform better biological understanding and better translate into biomedicine and clinical studies. Thus, the field of single molecule protein sequencing attempts to apply concepts from DNA and RNA sequencing to protein analyses in order to take advantage of the high parallelism, sensitivity, and throughput potentially offered by these approaches [1,2,3,4].

Fluorosequencing is one such single-molecule protein sequencing technique inspired by methods used for DNA and RNA [5,6]. In fluorosequencing, proteins in a biological sample are denatured and cleaved enzymatically into peptides. The researcher then chemically labels specific amino acid types, or alternatively, specific post-translational modifications (PTMs), within each peptide with different fluorescent dyes, then covalently attaches the peptides by their C-termini to the surface of a single-molecule microscope imaging flow-cell (Fig 1A). Sequencing proceeds by alternating between acquiring fluorescence microscopy images of the immobilized peptides and performing chemical removal of the N-terminal-most amino acid from each peptide, using the classic Edman degradation chemistry [7,8] (Fig 1B). In this manner, the sequencing cycle (corresponding to amino acid position) at which different fluorescent dyes are removed is measured on a molecule-by-molecule basis, with these data collected in parallel for all the peptide molecules observed in the experiment (Fig 1C).

In theory, this process gives a direct readout of each peptide's amino acid sequence, at least for the subset of labeled amino acids (Fig 1D), but in practice there are several complications because of the single-molecule nature of this sequencing method. Single molecule fluorescence intensities are intrinsically noisy, arising from the repeated stochastic transitions of each individual dye molecule between ground state and excited state, making stoichiometric data inexact, particularly when there are large fluorophore counts. Typically, no more than 5–6 copies of the same amino acid, hence dye, are expected for average proteolytic peptide lengths, with the number of distinct colors (*i.e.*, fluorescent channels) set by the microscopy optics and available dyes, here assumed to be 5 or fewer. However, inevitably with any chemical process, some fluorophore labeling reactions fail to occur, and photobleaching or chemical destruction can destroy fluorophores in the middle of a fluorosequencing run. At some low rate, peptides may detach from the flow-cell during sequencing, and Edman degradation can skip a cycle. These error rates, while individually small (approximately 5% each in published analyses [2]), collectively add difficulty to peptide identification, necessitating computational methods to process these data.

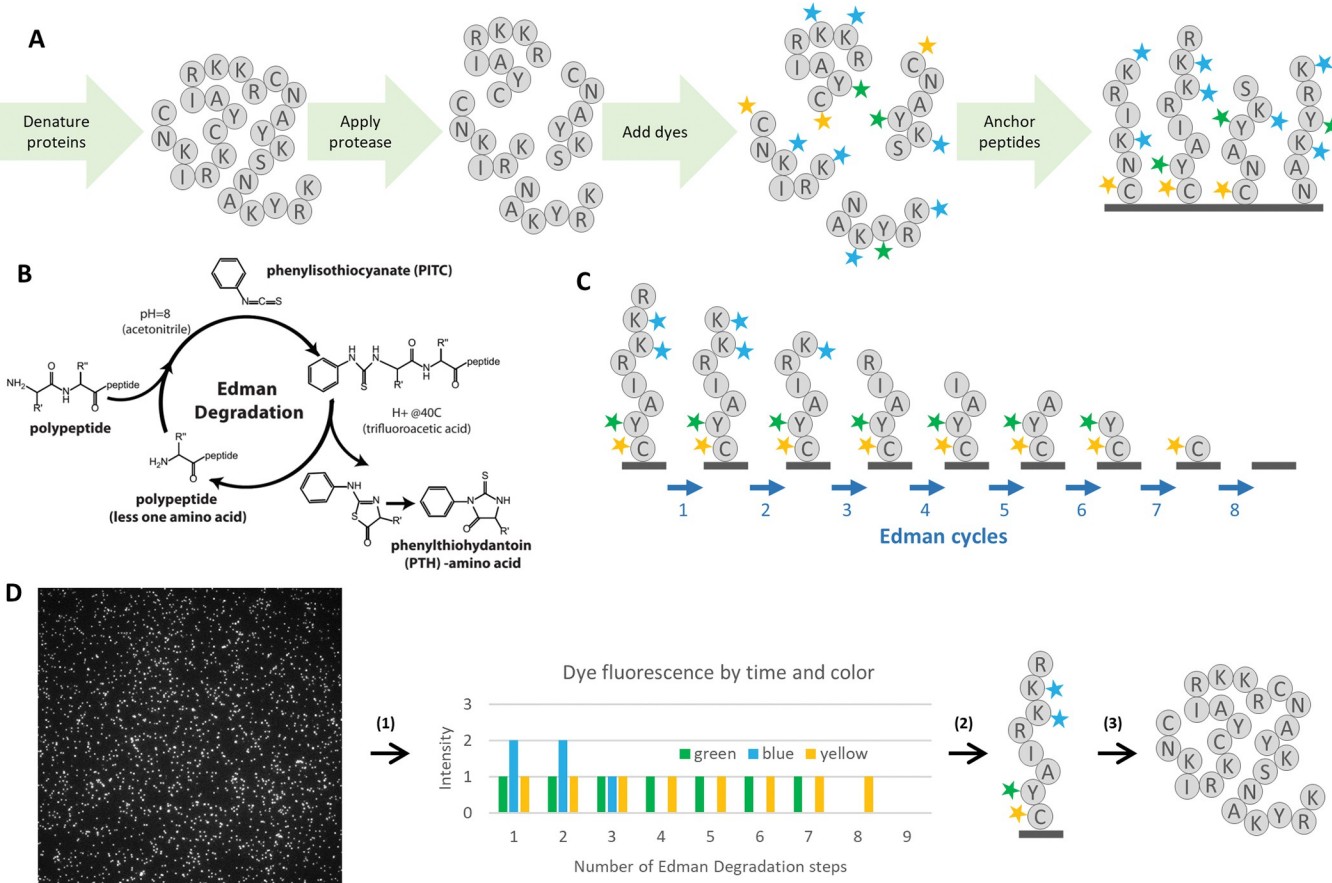

**Fig 1. Overview of protein fluorosequencing.** (A) illustration of the sample preparation process. Each grey circle represents an amino acid, and the letter in the circle corresponds to the standardized single letter amino acid codes. In the diagram, proteins are denatured, cleaved with protease, labeled with fluorescent dyes, and then labeled peptides are immobilized by their C-termini on the surface of a flow-cell. (B) The Edman degradation chemical reaction cycle, used to predictably remove one amino acid per cycle from each peptide. (C) For a given peptide, the sequencing process removes amino acids one at a time from the N-terminus, taking with them any attached fluorescent dyes. (D) Major steps in computational data analysis include: (1) For each field of view, performing image analysis to extract fluorescence intensities for each spot (peptide) in each fluorescent channel across time steps (cycles), collating the fluorescence intensity data per spot across timesteps and colors. A vector of fluorescence intensities is produced, giving a floating-point value for every timestep and fluorophore color combination. (2) These raw sequencing intensity vectors (raw reads) must then be classified as particular peptides from a reference database. This step is the primary concern of this paper. (3) Identified peptides can then be used to identify and quantify the proteins in the original biological sample.

Currently, there are no published algorithms for mapping fluorosequencing reads to a reference proteome to identify the proteins in a sample. The first analyses of fluorosequencing data used Monte Carlo simulations to generate realistic simulated data as a guide for data interpretation and fitting of experimental error rates [5,6]. While this strategy did not scale well computationally to full proteomes, it suggested that probabilistic modeling of the fluorosequencing process could provide a powerful strategy for interpreting these data. In this paper, we explore the application of machine learning to develop a classifier that correctly accounts for the characteristic fluorosequencing errors but is computationally efficient enough to scale to the full human proteome.

Viewing this as a machine learning problem is challenging due to the large numbers of possible peptides in many biological experiments. For example, in the human proteome, there are about 20,000 proteins, which when processed with an amino-acid specific protease such as trypsin can correspond to hundreds of thousands or even millions of distinct peptides, each of which can potentially vary due to post-translational modifications or experiment-specific

processing. This puts fluorosequencing data analysis squarely in the realm of Extreme Classification problems, which are known to be challenging to handle in practice [9].

To analyze these data, we took advantage of the ability to generate simulated fluorosequencing data using Monte Carlo simulations [5,6] to test k-Nearest Neighbors (kNN) classification and found it gave results of poor quality but is able to scale efficiently to the full human proteome while maintaining reasonable runtimes. These initial explorations motivated the developments presented in this manuscript, which focuses on the specific challenge of matching fluorosequencing reads to peptides from a reference proteome (*peptide-read matching*).

Here, we propose a specialized classifier which combines heavily optimized Hidden Markov Models (HMMs) to model the peptide chemical transformations during fluorosequencing, in combination with kNN pre-classification to reduce runtime. We call this tool *whatprot*, compare it with kNN and a classifier which uses HMMs without the kNN based runtime reduction, and demonstrate that the hybrid HMM-kNN approach offers a powerful and scalable approach for interpreting protein fluorosequencing data with the use of a reference proteome.

## Methods

### Monte Carlo simulation

To generate training and testing data typical of fluorosequencing experiments, we performed Monte Carlo simulations based on the model and parameters described in [5] [6]. These parameters are the dye loss rates $p_c$, which differ for each color $c$, the missing fluorophore rates $m_c$, the Edman cycle failure rate $e$, the peptide detachment rate $d$, the average fluorophore intensity $\mu_c$, and the standard deviation of fluorophore intensity $\sigma_c$. We additionally model a background standard deviation $\sigma'_c$. Based on prior estimates for the dye Atto647N ([5,6]), we used the following values unless otherwise noted: $p_c = .05$, $m_c = .07$, $e = .06$, $d = .05$, $\mu_c = 1.0$ (arbitrary rescaling of intensity values), $\sigma_c = .16$, $\sigma'_c = .00667$. Although the code permits different values for different colors $c$, for our simulations, we modeled each color of fluorophore with identical error values for simplicity.

An overview of the process with definitions for key terms is provided in Fig 2. We generate simulated data in two formats. The first of these formats we refer to as a *dye track*, and it indicates the number of remaining fluorophores of each color at each time step after considering sequencing errors. Thus, each copy of one particular peptide sequence may give rise to a different specific dye track in a sequencing experiment depending on the details of the labeling schemes and sequencing efficiencies. To simulate a dye track, we randomly alter (with a pseudo random number generator) a representation of a dye sequence in a series of timesteps, writing to memory the count of each color of fluorophore as we progress until we reach a preset number of timesteps. In this simulation, we initially remove fluorophores with a probability of $m_c$ before beginning sequencing. We then additionally perform a series of random events after logging fluorophore counts for each timestep: we remove the entire peptide and all fluorophores with a probability of $d$ to simulate peptide detachment from the flow cell, we remove the last amino acid (and any attached fluorophore) with a probability of $(1−e)$, and we remove each fluorophore with a probability of $p_c$, where $c$ is the color of the fluorophore, to simulate fluorophore destruction. We then log each fluorophore count into the dye track at every cycle.

The other format of data we consider is a *raw read*, which consists of radiometry data for each fluorescent color and Edman cycle. Raw reads result experimentally from signal processing and radiometry of the microscope imaging data from a fluorosequencing experiment. To simulate a raw read, we first simulate a dye track, and then we convert each fluorophore count into a floating point value indicating the fluorescent intensity. When we have a dye track entry indicating $\Lambda_c$ fluorophores for a given fluorophore color $c$, we sample a normal distribution

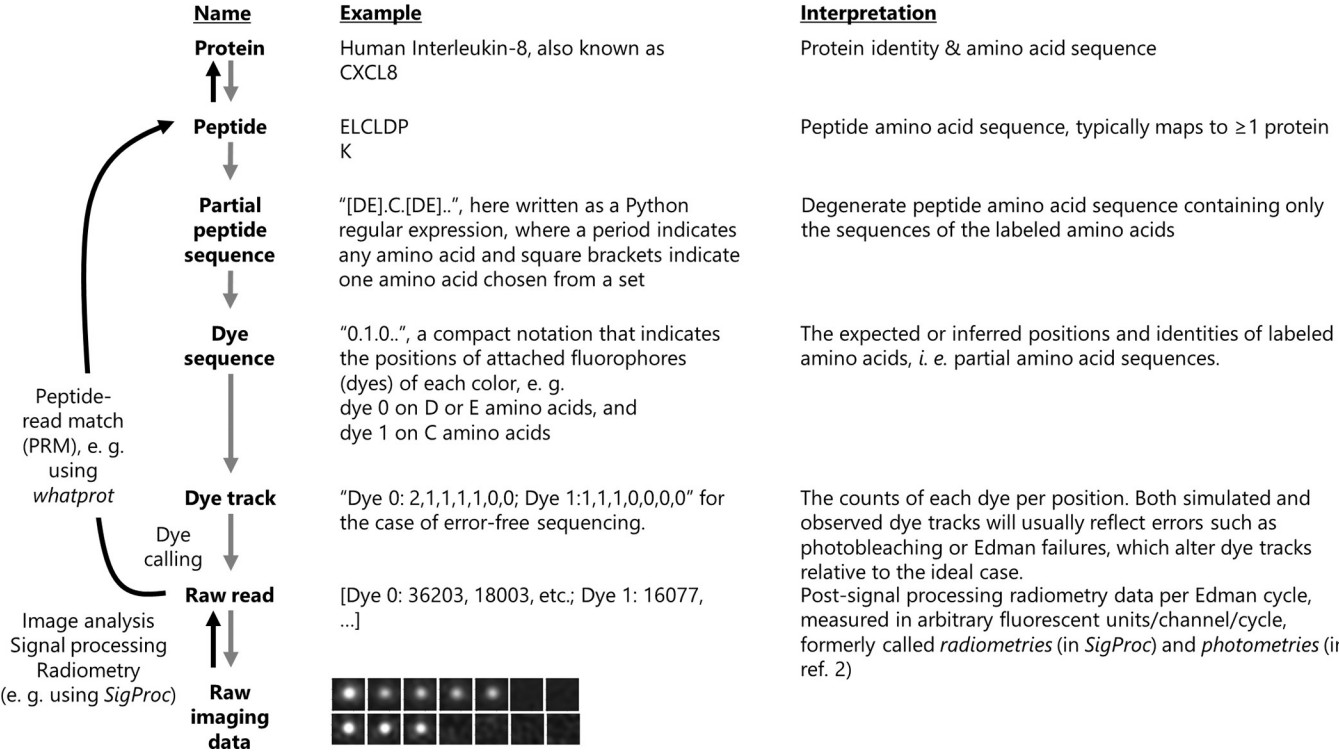

**Fig 2. Nomenclature for different stages of fluorosequencing data analysis.** The whatprot algorithm maps raw single-molecule protein sequencing reads to peptides and their parent proteins in the reference proteome (black arrows) by comparing experimental data (at bottom) to synthetic data generated using a Monte Carlo simulation (gray arrows).

with a mean of $\Lambda_c \mu_c$ and a variance of $\sigma_c'^2 + \Lambda_c \sigma_c^2$. We perform this calculation for each channel at each time-step to simulate a raw read.

These radiometry *raw reads* simulate the fluorescent intensity data we would expect to collect from processing raw single molecule microscope images, a process currently performed for experimental data using the algorithm *SigProc* (Part of Erisyon's tool *Plaster*, https://github.com/erisyon/plaster_v1), as in [10,11].

## Bayesian classification with HMMs

Whatprot builds an independent HMM for each peptide in a provided reference proteome dataset. Each state in this HMM represents a potential condition of the peptide, including the number of successfully removed amino acids, and the combination of fluorophores which have not yet photobleached or been destroyed by the chemical processing (Fig 3). Transition probabilities between these states can be calculated using previously estimated success and failure rates of each step of protein fluorosequencing.

We can use the HMM forward algorithm to associate a specific peptide to each *raw read* (a series of observed fluorescence intensities over time and across different fluorescence channels). We obtain the probability of the peptide given the raw read in two steps. First, we compute the HMM forward algorithm using each possible peptide in the dataset to obtain the probability of the raw read given each peptide. This uses the forward algorithm formula

$$\boldsymbol{f}^{(t+1)} = \boldsymbol{O}^{(t+1)} \boldsymbol{T} \boldsymbol{f}^{(t)} \tag{1}$$

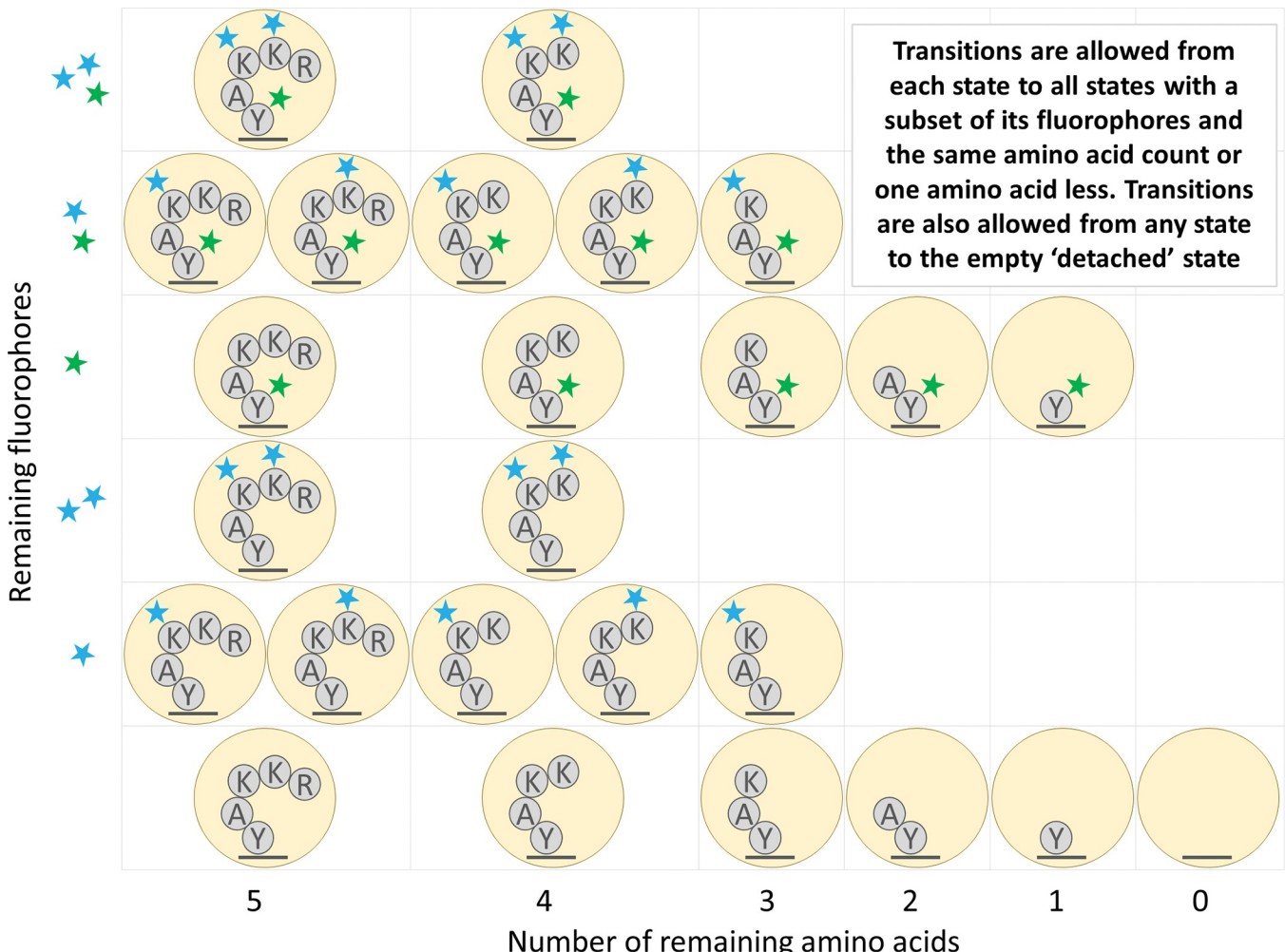

**Fig 3. Illustration of the states and transitions of the HMM for an example peptide.** For the amino acid sequence RKKAY, we illustrate the case where the lysine (K) residues are labeled with fluorescent dyes of one color (blue stars) and the tyrosine (Y) residue is labeled by a second color (green star).

Where $\boldsymbol{f}^{(t)}$ represents the cumulative probabilities for each state at timestep $t$, where $0 \leq t \leq T$, $\boldsymbol{O}^{(t)}$ represents the diagonal emission matrix for the observation seen at timestep $t$, and $\boldsymbol{T}$ represents the transition matrix which is the same at every timestep. The entries in each $\boldsymbol{f}^{(t)}$, $\boldsymbol{O}^{(t)}$, and in $\boldsymbol{T}$, represent the following probability densities:

$$\boldsymbol{f}_i^{(t)} = p(Y_{1:t} = y_{1:t}, X_t = i | Z = z) \tag{2}$$

Where $Y_{1:T}$ are the random variables for the observations, $y_{1:T}$ are their true values, $X_{1:T}$ are the random variables for the state in the HMM and $Z$ is the random variable representing the peptide, and $z$ is a value it can take. We also have diagonal matrices $\boldsymbol{O}^{(t)}$ defined as:

$$\boldsymbol{O}_{ii}^{(t)} = p(Y_t = y_t | X_t = i, Z = z) \tag{3}$$

And:

$$\boldsymbol{T}_{ij} = p(X_{t+1} = i | X_t = j, Z = z) \tag{4}$$

We start from an initial state $f^{(0)}$ which we compute by taking into account the missing fluorophore rate $m_c$. Applying (1) repeatedly starting with the initial state $f^{(0)}$ yields a value for $f^{(T)}$, and we can sum the entries to compute:

$$p(Y_{1:T} = y_{1:T}|Z = z) = \sum_i p(Y_{1:T} = y_{1:T}, X_T = i|Z = z) = \sum_i f_i^{(T)} \tag{5}$$

Then, by using Bayesian inversion to normalize the data, we compute the probability of the peptide given the raw read, as given by:

$$p(Z = z|Y_{1:T}p = y_{1:T}) = \frac{p(Y_{1:T}|Z = z)p(Z = z)}{\sum_{\tilde{z}} p(Y_{1:T} = y_{1:T}|Z = \tilde{z})p(Z = \tilde{z})} \tag{6}$$

We implemented several algorithmic optimizations to this approach to reduce runtime. These included reducing the number of states in the HMMs, factoring the HMMs' transition matrices into a product of matrices with higher sparsity, pruning the HMM forward algorithm to consider only reasonably likely states at each timestep, and combining the HMM classifier with a kNN pre-filter that can rapidly select a short-list of candidate peptides for re-scoring by the HMM. We implemented the linear algebra and tensor operations being performed in a manner that makes productive use of spatial and temporal locality of reference. We describe these optimizations in more detail in the following sections and in S1 Appendix.

## HMM state space reduction

We reduced certain physical states of a peptide into a single modeled state. We consider the physical state to be the fully specified physical arrangement of fluorophores on the peptide molecule being analyzed. In particular, in the physical state fluorophores are unambiguously attached to specific amino acids. However, the model we used allows multiple physical states to be combined into a single modeled state. In the modeled state, the locations of the fluorophores on the peptide are not explicitly defined, and we track only the counts of each color of fluorophore and the number of amino acids on the peptide. An example of the resulting HMM for a sample peptide is shown in Fig 4.

A similar state reduction to ours was previously described by Messina and colleagues in [12]. The reduction requires fluorophores to behave independently of each other, so that the status of one fluorophore is uncorrelated with the status of any other. While this is not true in practice due to FRET (Förster resonance energy transfer) and other dye-dye interactions, quantification of this effect in the imaging conditions used for fluorosequencing suggests that these effects are negligible enough to ignore [6]. The authors of [12] also require the fluorophores to be indistinguishable to reduce the numbers of states. This is not true in our case because we use Edman degradation and because we use multiple colors of fluorophores.

Nonetheless, we demonstrate in S1 Appendix that despite these complications, this state space reduction incurs no loss in the theoretical accuracy of the model. Further, we demonstrate that this reduces the algorithmic complexity from what would otherwise be exponential with respect to the number of fluorophores, to instead be tied to the product of the counts of fluorophores of each color.

## Transition matrix factoring

In the HMM forward algorithm, a vector of probabilities with one value for each state in the HMM's state space is repeatedly multiplied by a square transition matrix. This operation is the dominant contribution to the algorithmic complexity of the HMM forward algorithm.

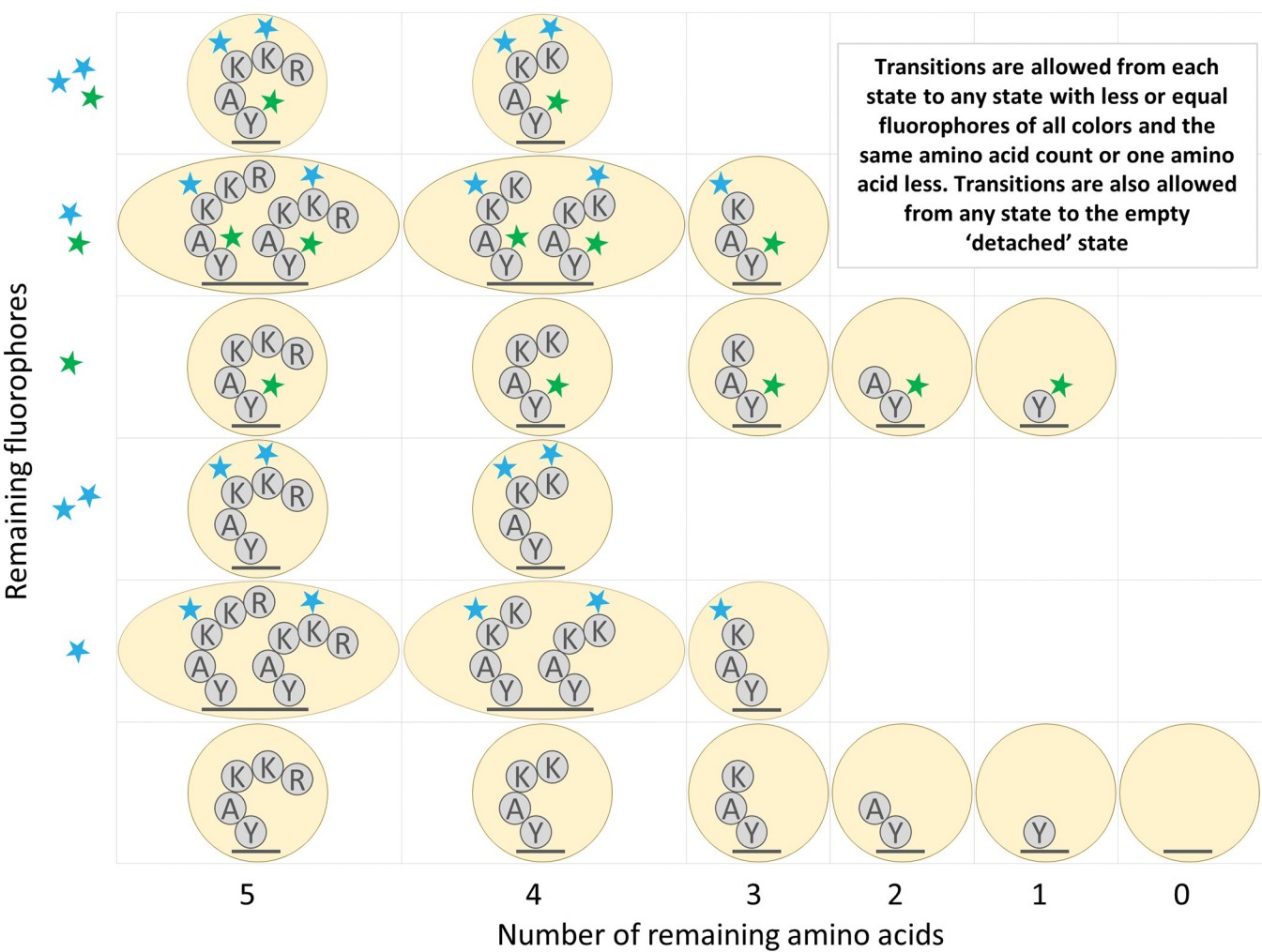

**Fig 4. Illustration of HMM state space reduction for the peptide of Fig 3.** States are combined that have both the same number of amino acids remaining and the same fluorophore counts for each color of fluorophore.

Therefore, by making multiplication by the transition matrix more algorithmically efficient, we can improve the algorithmic complexity of our computational pipeline.

We factor this transition matrix into a product of highly sparse matrices. This factorization is done by creating a separate matrix for each independent effect under consideration, including loss of each color of dye (where each color is factored separately), Edman degradation, and finally, peptide detachment (Fig 5). As with the state space reduction, this optimization incurs no loss in the accuracy of the model, and furthermore, these matrix factors, even in combination, are far sparser than the original transition matrix when computed for larger peptides. This greater sparsity can be leveraged to achieve superior algorithmic complexity results (see S2 Appendix).

## HMM pruning

Despite significant improvements in the algorithmic complexity of an HMM for one peptide given so far from state space reduction and matrix factorization, performance can be improved if we consider approximations. Intermediate computations contain mostly values close to zero, which will have inconsequential impact on the result of the HMM forward algorithm.

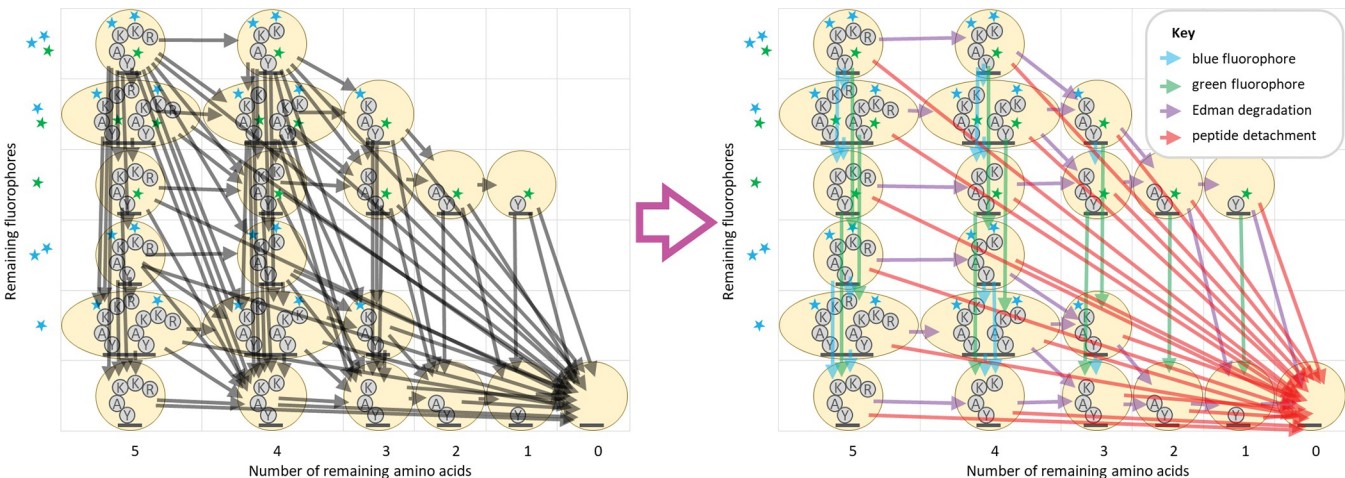

**Fig 5. An illustration of the factoring of the transition matrix for the peptide from Fig 4.** Note especially the reduction in the total number of transitions (arrows) when the transition matrix is factored. At left, black arrows represent non-zero entries in the unfactored transition matrix. At right, colored arrows (see key) represent non-zero entries in each of the matrices in the factored product. In both diagrams, arrows from a state to itself are omitted for visual clarity.

The most significant contribution to this occurs for the HMM emission calculations. While there may be many states of a peptide which have a significant probability of producing a particular observation value, in most states (particularly for larger peptides) the observed value is extremely unlikely.

Emission computations can be viewed as multiplication by a diagonal matrix, different for each emission in a raw read. The entries represent the probability of the indexed state producing the known emission value for that timestep. We prune this matrix by setting anything below a threshold to zero, which increases the sparsity of the matrix. Although use of a *naïve*, but standard, sparse matrix computational scheme would reduce runtime, we show that better algorithmic complexity can be achieved with a more complicated bi-directional approach in Fig 6 and S3 Appendix. While we did not implement this approach precisely, a consideration of this effect served as inspiration for a technique combining pruning with matrix factoring, as described next.

## Combining transition matrix factoring with HMM pruning

Both transition matrix factoring and HMM pruning appear, at first glance, to be incompatible improvements. These approaches can be combined, but the bi-directional sparse matrix computational scheme introduces significant additional difficulties.

We view the various factored matrices as tensors and propagate contiguous blocks of indices forwards and backwards before running the actual tensor operations to avoid unnecessary calculations (Fig 7). Contiguous blocks of indices are needed because propagating lists of indices across the various factors of the matrices has the same computational complexity as multiplying a vector by these matrices. This may make the pruning operation seem less optimal in a sense, as some values that get pruned may be bigger than some that are kept due to this form of indexing. Nevertheless, we found the tradeoff to be favorable in practice (S4 Appendix).

Of note, instead of pruning by the raw values, we prune all states such that the known emission value is outside of their pre-configured confidence interval. The bounds of this confidence interval is determined by a cut-off value that is multiplied by $\sigma_c$ similarly to what you might do with a z-score. In this way we provide some confidence that the fraction of true data inadvertently zeroed out is negligible.

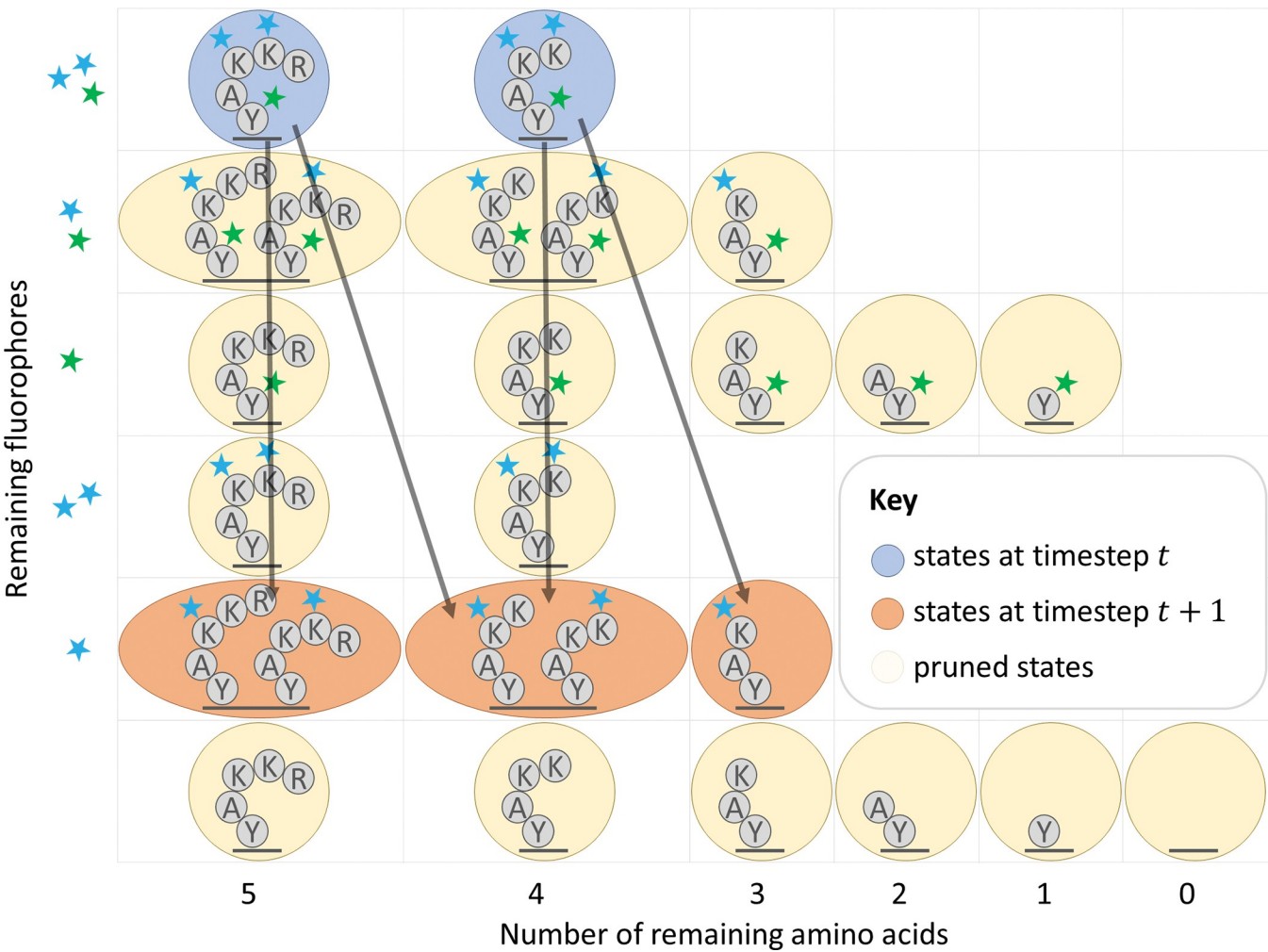

**Fig 6. Illustration of the effects of HMM pruning for the peptide of Fig 5.**

### *k*-Nearest Neighbors classification

Most traditional machine learning classifiers have an algorithmic complexity which scales proportionally or worse to the number of classification categories. The Bayesian classifier we have so far described is no exception; each raw read must be compared against every peptide in the reference dataset to be classified. There are many problems in biology which require large reference datasets, human proteomic analysis being one example. The human proteome has 20,000 proteins, which when trypsinized generate hundreds of thousands of peptides. Classification against these many categories is computationally intractable with a fully Bayesian approach.

In contrast, the algorithmic complexity of kNN scales logarithmically with the number of training points used. For this reason, tree-based methods are common in other Extreme Classification applications [9], where similarly massive numbers of categories are under consideration. Unfortunately, the resulting faster runtimes come at a significant cost; kNN often gives far worse results in practice than a more rigorous Bayesian approach.

For purely kNN based classification, we simulate 1000 raw reads per peptide in the reference to create a training dataset and put these into a custom KD-Tree implementation for fast

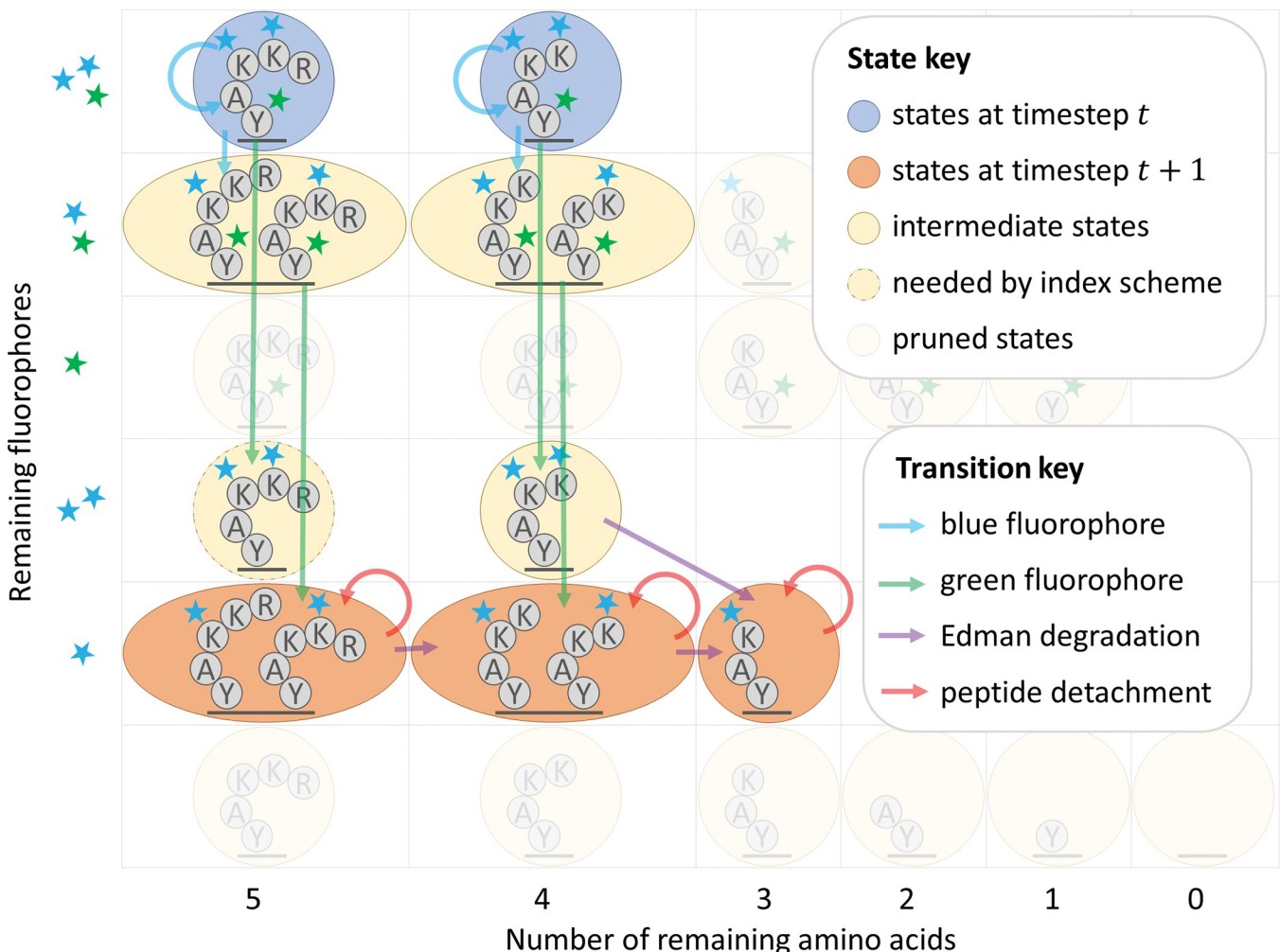

**Fig 7. Illustration of HMM pruning combined with transition matrix factoring for the peptide of Fig 5.** We emphasize that this is an anecdotal example; while there are more arrows here than in Fig 6, this strategy provides an improvement in asymptotic complexity, as described in S4 Appendix and shown in experiments with simulated data.

and easily parallelizable nearest neighbor lookups. We do not allow edits in our KD-Tree after it is built so as to allow parallelized lookups to occur without any concern for locks or other common issues in parallel data structures. We also reduce the memory footprint of the KD-Tree through an unusual compression scheme. For our training data, we use dye tracks instead of raw read radiometry data; this alone reduces the memory footprint of the KD-Tree by a factor of four due to the less precise format. But this allows another further compression technique; we find all identical dye tracks and merge them into one entry. With these dye tracks entries in the KD-Tree we store lists of peptides that produced the dye track when we simulated our training data, along with how many times each peptide produced that dye track.

To classify an unknown raw read, the $k$ nearest dye track neighbors to a raw read query are retrieved, using a Euclidean norm as the metric. These neighbors then vote on a classification, with votes weighted using a Gaussian kernel function, $\exp\left(-\frac{\delta^2}{2\sigma_{kNN}^2}\right)$, where $\delta$ is the Euclidean distance between the query raw read and the neighbor, and $\sigma_{kNN}$ is a parameter of the algorithm. A neighbor is also weighted proportionally to the number of times it occurred as a

simulation result and will split its voting weight among all of the peptides that produced that dye track proportionally to the numbers of times each peptide produced the dye track during simulation of training data.

Once voting is complete, the highest weighted peptide is then selected as the classification, with its classification score given as a fraction of its raw score over the total of all the raw scores. We have explored multiple choices of $k$ and $\sigma$ values to optimize the performance.

### Hybridizing kNN with Bayesian HMM classification

To combine the computational efficiency of kNN with the accuracy of the HMM model, we defined a classifier which hybridizes these two disparate methods. We use a kNN classifier to reduce the reference dataset, for each raw read, down to a smaller shortlist of candidate peptides. These candidates can then be used in the Bayesian classifier by building HMMs to compare them against the specific raw read. While this can result in the true most likely peptide not being in the shortlist and therefore not being selected by this hybrid classifier, with a sufficiently long shortlist this is highly unlikely. A larger problem is in performing Bayes' rule, as in (6). An exact formula for Bayes' rule requires an exhaustive set of probability values for every potential outcome, which are summed in the denominator. Avoiding determining every probability makes this impossible. Instead, we can estimate Bayes' rule as follows:

$$p(Z = z | Y_{1:T} p = y_{1:T}) = \frac{p(Y_{1:T} | Z = z) p(Z = z)}{\sum_{\tilde{z} \in \zeta_h} p(Y_{1:T} = y_{1:T} | Z = \tilde{z}) p(Z = \tilde{z})}$$

Where $\zeta_h$ is the set of up to $h$ peptides selected by the kNN method; we require $z \in \zeta_h$. Although we lose theoretical guarantees of optimal accuracy given the model, this change provides a considerable improvement to the algorithmic complexity. The algorithmic complexity to classify one raw read using a fully Bayesian approach is $O(RW)$, where $R$ is the number of peptides in the reference dataset, and $W$ is the average amount of work needed to run an HMM for one peptide fluorophore combination. In comparison, with the hybridized classifier, the algorithmic complexity is $O(\log(RQ) + hW)$ where $Q$ is the number of raw reads in the training dataset simulated for each possible peptide.

We chose specific values for $h$, $\sigma$, and $k$, by comparing the runtime and PR curves on simulated datasets.

### Maintaining spatial locality of reference

Spatial and temporal locality of reference is the tendency of some computer programs to access nearby data points at similar times. Modern CPUs are designed to make this extremely efficient through multi-level batch caching schemes which cache data from RAM that is nearby a memory address being accessed, so that nearby data can be read more quickly. Programs which exploit this in read-write intensive pieces of code can often achieve significant runtime acceleration compared to programs which do not.

We wrote highly optimized kernel functions to perform our structured matrix/tensor operations which exploited the sparse nature of the problem while also iterating over elements in what we believe to be an optimal or near optimal fashion for most computer architectures. We believe this provided considerable improvements in performance, though this has not been rigorously tested.

## Results

We simulated the fluorosequencing of peptides to obtain labeled training and testing data (Fig 2). We generated several datasets, each with a randomized subset of the proteins in the human

proteome. We selected 20 proteins (.1% of the human proteome), 206 proteins (1% of human proteome), 2,065 proteins (10%), and 20,659 proteins (the full human proteome). We repeated this randomized selection scheme to examine several protease and labeling schemes. These were (1) trypsin (which cleaves after lysine (K) and arginine (R) amino acids) with fluorescent labels for aspartate (D) and glutamate (E) (these share a fluorophore color due to their equivalent reactivities), cysteine (C), and tyrosine (Y), (2) cyanogen bromide (which cleaves after methionine (M) amino acids) with D/E, C, Y, and K, (3) EndoPRO protease (which cleaves after alanine (A) and proline (P) amino acids) with D/E, C, and Y, (4) EndoPRO with D/E, C, Y, and K, (5) EndoPRO with D/E, C, Y, K, and histidine (H). Thus, in the schemes examined, of the 20 canonical amino acid types found in most proteins, either one or two were recognized by the protease and up to 6 additional amino acids were labeled by fluorescent dyes.

These databases of peptides were used to generate databases of idealized *dye sequences*, *dye tracks*, and *raw reads*, used for training and testing purposes for the various models. Peptides were converted directly into dye-sequences and combined as appropriate. For our training data for kNN, we then generated 1000 dye-tracks for each peptide in the dataset. We also note that the Bayesian HMM classifier requires no training data, as it is based on a direct physical model of the fluorosequencing process.

For our test data for each dataset, we generated 10,000 raw sequencing reads by randomly selecting peptides with replacement from the dataset and simulating sequencing on them using the techniques described in the Monte Carlo simulation Methods section. For both dye tracks and simulated fluorescent intensity measurements, results where there were zero fluorophores throughout sequencing were discarded, as these would fail to be observed in an actual sequencing experiment.

All timed runs were executed on the Texas Advanced Computing Center (TACC) at The University of Texas at Austin. We used their stampede2 system, and in particular their standard Intel Xeon Platinum 8160, or "Skylake," nodes. These nodes had 48 cores split between two sockets (24 cores/socket), with 2 hardware threads per core, giving 96 hardware threads on a node. The nodes we used also had 192GB of DDR4 RAM.

We collected and compared runtime data and precision-recall curves for several different purposes. With the trypsinized 3-color dataset, we performed a parameter sweep of the pruning cut-off for the HMM Bayesian classifier (S1 Fig). Losses in precision and recall performance were negligible for cut-off values of 5 and greater, though the precision recall curves grew worse at smaller values. Runtimes shrank rapidly as the cut-offs were decreased. The pruning cut-off parameter sweep was also performed on the 20-protein cyanogen bromide dataset (S2 Fig). We saw that in this second dataset, runtime improvements for lower cut-offs were even more extreme; a speed-up factor of about 1000 could be achieved with minimal effects on the precision recall plots. From these two simulations, we chose a cutoff value of 5 as providing the optimal trade-off between runtime and precision recall performance.

On the trypsinized dataset (full human proteome), we also swept the $k$ and $\sigma_{kNN}$ parameters of the NN classifier (S3 and S4 Figs). Here large values of $k$ introduce modest reductions in precision recall performance, while the model is extremely sensitive to the selection of $\sigma_{kNN}$. Based on this analysis, we suggest that good choices of these parameters are $k = 10$ and $\sigma_{kNN} = 0.5$.

We swept all parameters of the hybrid classifier for the trypsinized dataset as well (hybrid $h$ parameter, $k$, $\sigma_{kNN}$, and cut-off) (S5–S8 Figs). Here, the HMM cut-off parameter had less impact on runtime than for the pure HMM Bayesian classifier, but we still found a cut-off of 5 to be optimal. Higher values of $k$ improved precision recall performance for the hybrid model, contrary to the results of the NN classifier on its own, and we therefore suggest setting $k$ to 10000. $\sigma_{kNN}$ had minimal impact of any kind, in contrast to its significant impact on the

precision recall of the NN classifier; we nevertheless chose to set it to 0.5 in light of the data from parameter tuning for the NN classifier on its own. We also found that higher values of $h$ improved performance, though the impact plateaus after $h$ of about 1000, and we used that value for later experiments.

After tuning parameters, we compared the performance of the different classifiers when applied to 10,000 simulated fluorosequencing reads of peptides drawn randomly from all tryptic peptides in the human proteome (Fig 8). The hybrid classifier achieved similar precision recall curves to the Bayesian HMM classifier, which was much better than the precision recall curve of the NN classifier. The hybrid classifier also achieved runtimes of the same order of magnitude as the NN classifier, which was significantly faster than the runtime of the Bayesian HMM classifier.

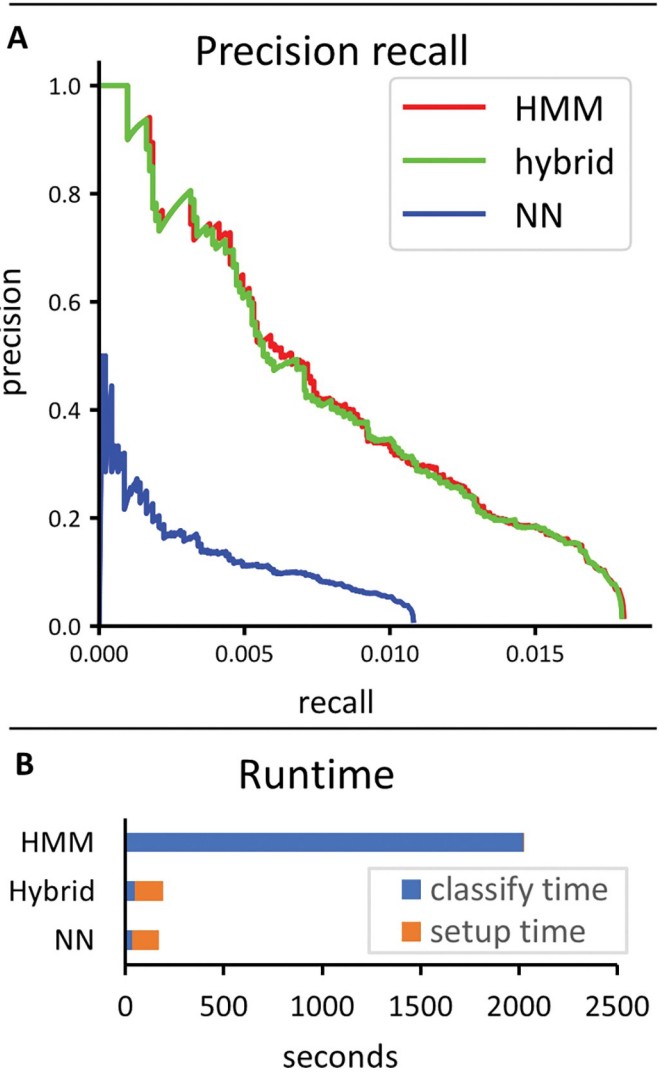

**Fig 8. Comparison of the HMM (Bayesian), hybrid, and NN classifiers.** This was done on a dataset of 10K reads from peptides chosen randomly from all 20,659 human proteins trypsinized and labeled on D/E, C, and Y. (A) The precision recall curves. (B) Runtimes.

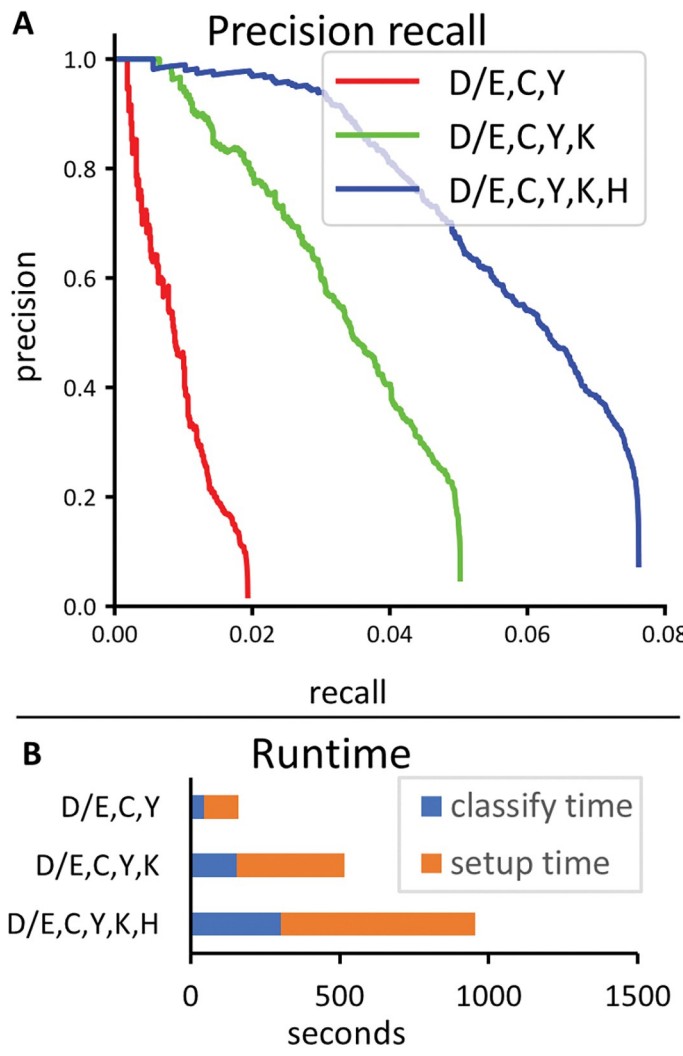

**Fig 9. Comparison of different labeling strategies.** The hybrid classifier was run on a dataset of 10K reads from peptides chosen randomly from all 20,659 human proteins cleaved with EndoPRO. (A) The precision recall curves. (B) Runtimes.

We also studied how the number of fluorophore colors affected the runtime and precision/recall of the hybrid classifier (Fig 9). We found that improvements in precision recall were possible with each additional color of fluorophore, but this did come at the cost of longer runtimes.

We also investigated the effect of varying sizes of reference proteomes on the hybrid classifier's performance, using the three color trypsinized dataset (Fig 10). We found that significantly better performance was possible when the reference database was smaller.

The precision/recall curves plotted above (Figs 8–10) show the actual precision/recall scores based on data with known peptide classifications. When working with real data this will typically not be possible, because the real classifications will not be known. It is therefore important that the assignment probabilities produced by the classifier be well-calibrated, so that an estimate of the precision/recall (or as is more often the case in protein mass spectrometry, the

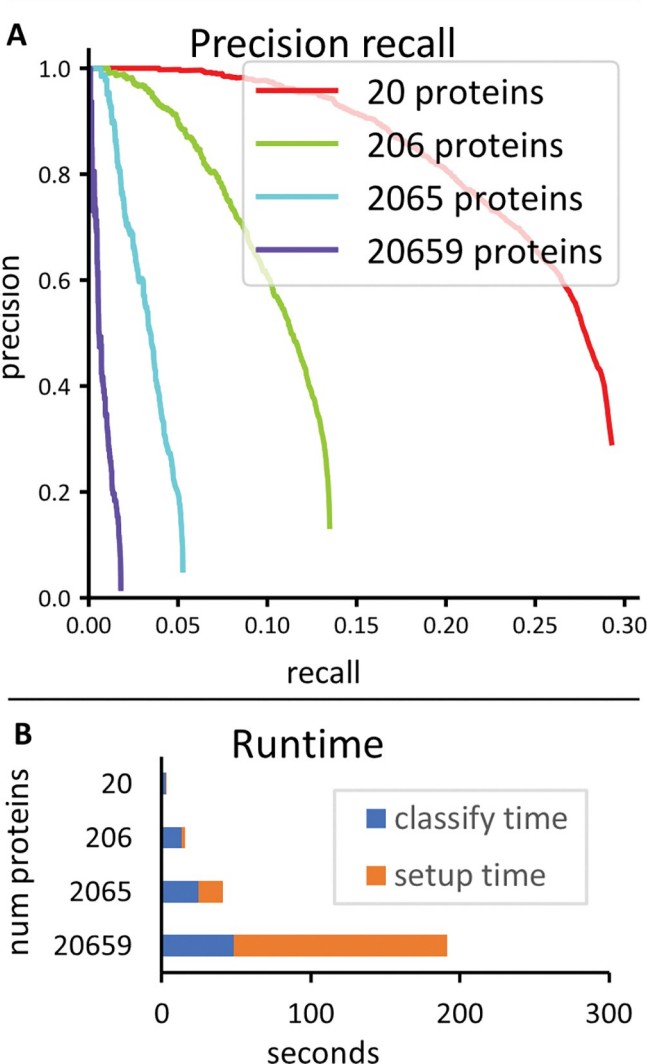

**Fig 10. Comparison of size of reference database.** The hybrid classifier was run on datasets of 10K reads each from peptides chosen randomly from different numbers of random human proteins treated with the same protease and labeling scheme (trypsin, labeled D/E,C,Y). (A) The precision recall curves. (B) Runtimes.

false discovery rate (FDR)) can be computed in the absence of known labels. We verified that the probabilities output by the hybrid HMM classifier were indeed well-calibrated relative to the true assignment probabilities (Fig 11A). This in turn allowed us to compute a predicted precision/recall curve assuming that each classification is fractionally correct with a probability given by its classification score. A comparison of this predicted P/R with the actual precision/ recall curve for the same set of reads shows excellent agreement (Fig 11B).

Whatprot specifically attempts to assign each raw fluorosequencing read to one or more peptides from the reference database, *i.e.*, to identify and score *peptide-read matches* (PRMs), a process highly analogous to analytical interpretation of shotgun mass spectrometry (MS) pro-teomics data in which a key step is comparing experimental peptide mass spectra to a reference

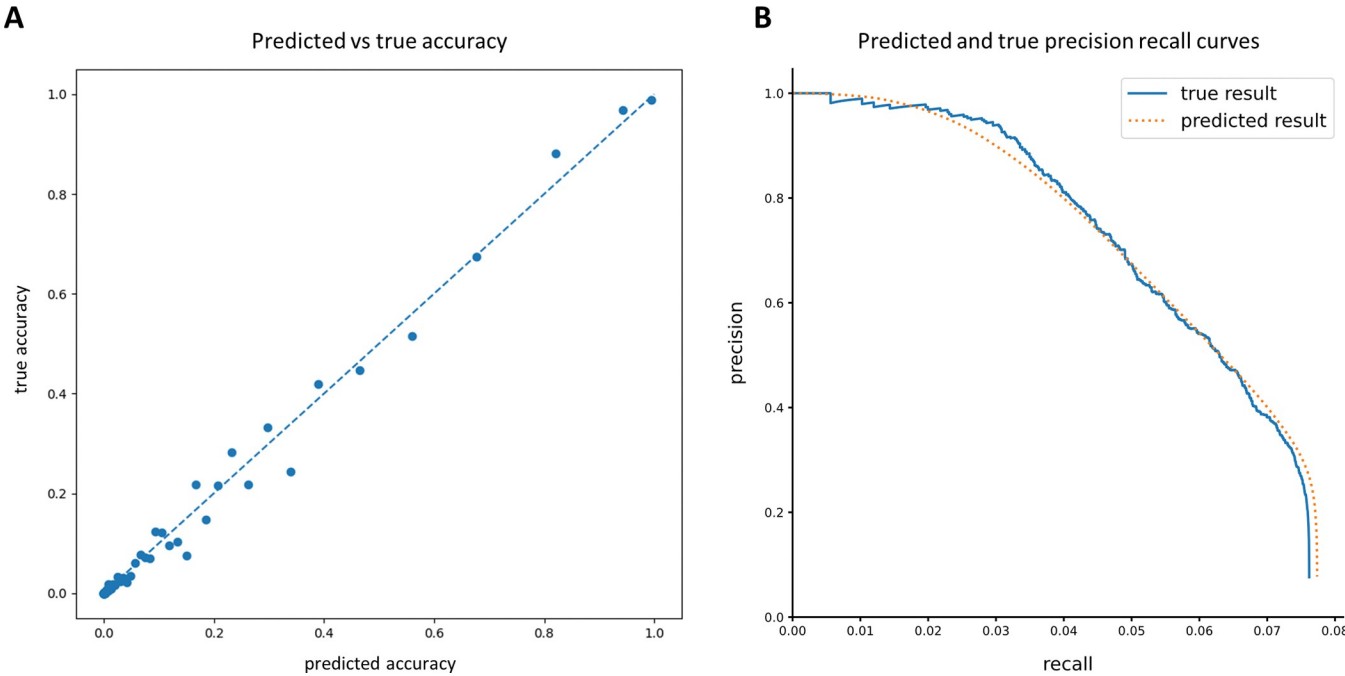

**Fig 11. Analysis of the accuracy of probability estimates given as scores by the classifier.** Based on 10K reads from peptides chosen randomly from all 20,659 human proteins cleaved with EndoPRO and labeled on D/E,C,Y,K,H. (A) Classification results were sorted by their predicted accuracy scores, and then equally distributed between 100 buckets. The average predicted and true accuracy scores were then computed for each bucket and plotted. (B) The true result precision/recall curve was computed as normal, while the predicted result precision/recall curve was plotted assuming each classification was fractionally correct according to its predicted accuracy score.

proteome (finding *peptide-spectral matches*, or PSMs [13]). However, observing multiple reads mapping to the same peptide will tend to increase the confidence that peptide is present in the sample, just as observing multiple peptides from the same protein will similarly increase confidence in that protein being present. Thus, we asked if considering the PRMs collectively led to performance increases for identifying peptides and proteins.

As shown in (Fig 12), proteins can be identified correctly at much higher rates than peptides, which are similarly identified at higher rates than individual reads. In fact, provided that a protein possesses some well-identified peptides that are unique, it can typically be identified with very high accuracy. For this test, we used a very simple protein inference scheme. First each peptide was scored to the maximum score of all reads identifying it, while penalizing reads which identified more than one peptide (dividing by $n$ if $n$ peptides were identified). Second each protein was scored as the maximum score of all peptides it contains, penalizing peptides which are associated with more than one protein (again dividing by $n$ if $n$ proteins were associated). However, the problem of integrating peptide level observations to protein observations has been studied extensively for MS [14,15], and it is likely that these techniques will offer similarly strong interpretive power to the case of single molecule protein sequencing.

We also wished to explore whether the parameterization of our model may have been overfit to human data. Though our parameter sweeps in S1–S8 Figs were coarse, and the parameters are general in a way that seems unlikely to have species specific effects, we confirmed overall performance by testing the same parameter choices against additional sets of proteins selected from other organisms. We tested precision and recall at the read, peptide, and protein levels for both *C. elegans* (Fig 13), and yeast (Fig 14). The results indicate that the tuning of

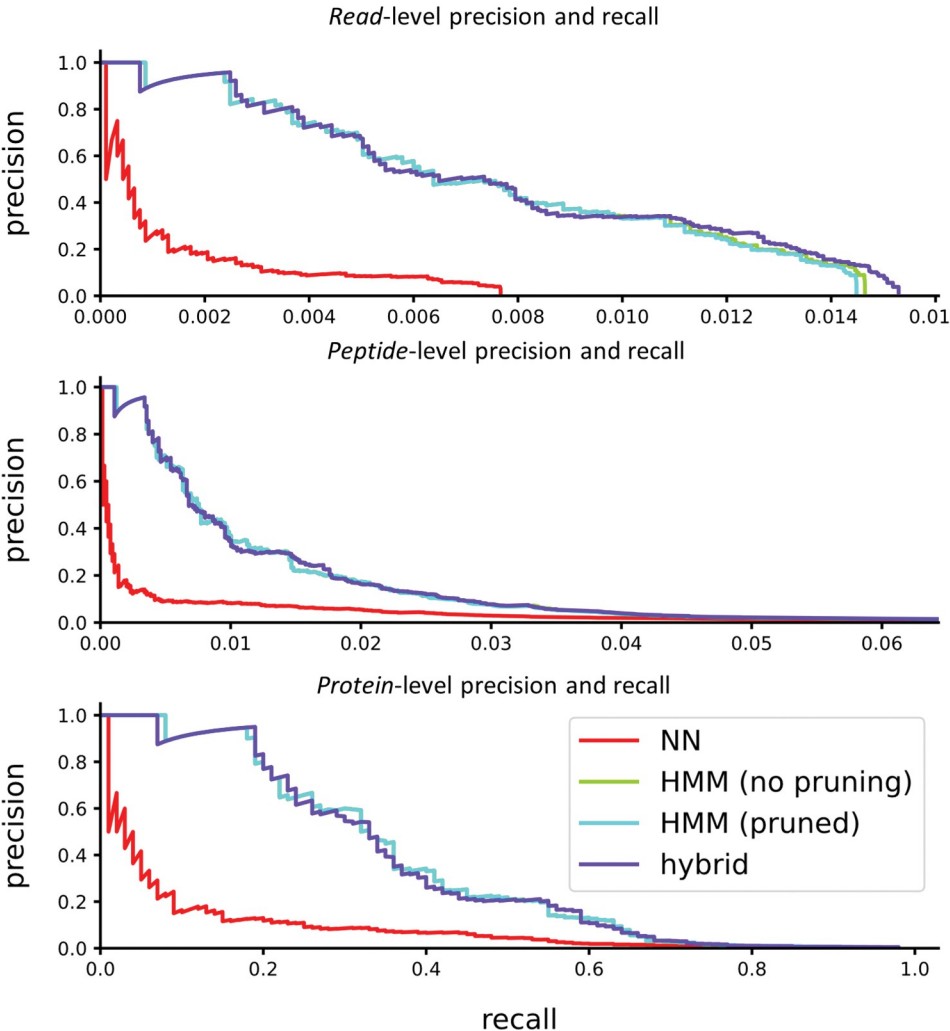

**Fig 12. Precision and recall are improved for proteins by integrating identifications across peptides.** The example shows 10K simulated reads from peptides derived from 100 proteins randomly selected from the human proteome, considering trypsin digestion and labels on D/E, C, and Y. We note the "HMM (no pruning)" curve is hidden under the "HMM (pruned)" curve.

parameters allows HMM pruning and a hybrid approach to accurately estimate the true best-estimate as given by an unpruned HMM-Bayesian classifier, even for non-human organisms.

We also explored the sensitivity of our parameterization to differences between the model parameters and the true data error rates. We applied the same optimized model to synthetic datasets generated with different error rates (e.g. Edman degradation rate, bleach rates, etc.), plotting the results in S9–S15 Figs. In general, the algorithm is reasonably robust to variation in the error rates, with some (such as mu) more sensitive than others (such as sigma). Not surprisingly, high true error rates degraded model performance, while lower than expected error rates led to improved performance in multiple cases. Thus, for several parameters performance was more adversely affected by higher error rates in the data than by differences between the model and the data, such as is evident for the dud-dye rate.

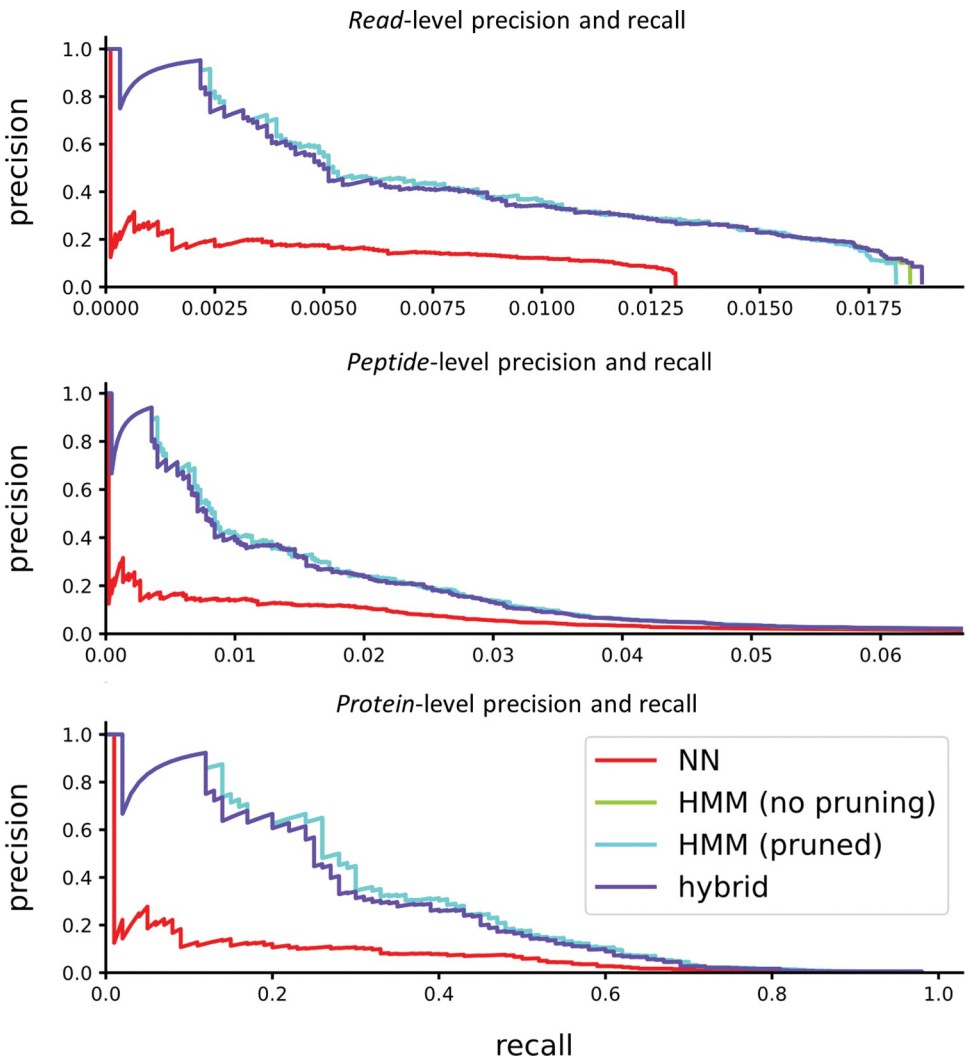

**Fig 13. Performance on *C. elegans*.** The example shows 10K simulated reads from peptides derived from 100 proteins randomly selected from the *C. elegans* proteome, considering trypsin digestion and labels on D/E, C, and Y. We note the "HMM (no pruning)" curve is hidden under the "HMM (pruned)" curve.

## Discussion

We developed an HMM for interpreting single molecule protein fluorosequencing data and showed that a hybrid HMM/kNN model can achieve a high precision and recall comparable the HMM alone while maintaining a runtime comparable to the much faster kNN.

It is worth emphasizing that these analyses were performed on datasets of 10,000 raw fluorosequencing reads. In practice, users will likely want to analyze millions to billions of reads, so runs that completed in a seemingly reasonable amount of time might still be intractable in these scenarios with larger datasets, or at a minimum require computing clusters with high parallelization. For analyzing the current datasets in the runtime charts (Figs 8–10), note that the *blue* part of the bar graphs indicates the classify time, which will scale with the number of reads being classified (if all else remains equal), and the *orange* part of the graph indicates setup time, which should remain constant regardless of the number of reads (though it changes depending on the model and the size of the reference set).

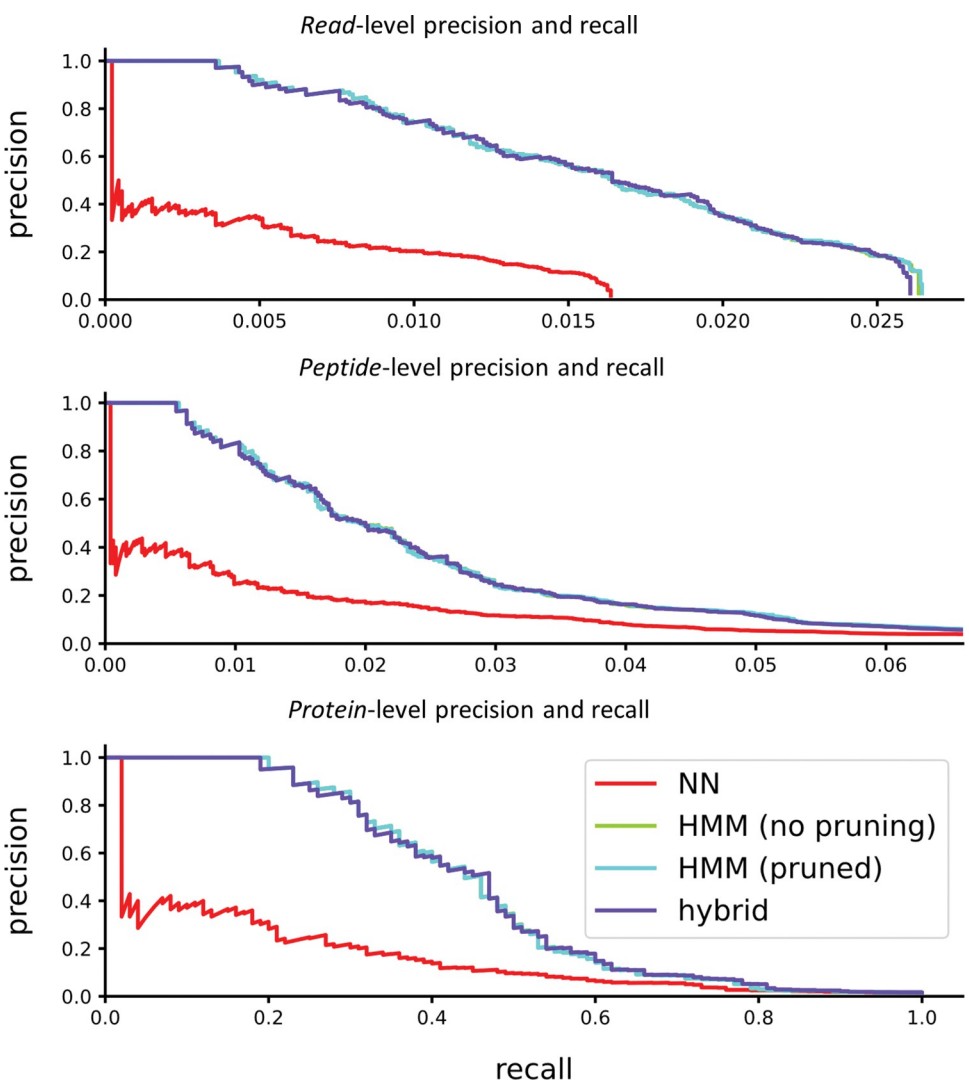

**Fig 14. Performance on yeast.** The example shows 10K simulated reads from peptides derived from 100 proteins randomly selected from a yeast proteome, considering trypsin digestion and labels on D/E, C, and Y. We note the "HMM (no pruning)" curve is hidden under the "HMM (pruned)" curve.

It is also interesting that the HMM pruning operation is more necessary with longer peptides and more colors of fluorophores; with the trypsinized dataset labeling D/E, C, and Y, omitting pruning had little consequence, but in moving to cyanogen bromide with D/E, C, Y, and K, we observed a runtime speedup of about 1000-fold. The algorithmic complexity without HMM pruning is tied to the number of states in the model, because each state must be visited at every step of the forward algorithm. The number of states in the model grows with the product of the numbers of fluorophores of each color, and pruning improves runtime by restricting to narrower ranges of fluorophore counts that the forward algorithm needs to consider at each timestep. This effect is multiplicative in the number of fluorophore colors, which we believe explains most of the improvement between these two scenarios. Longer peptides will typically have more labelable amino acids, which could amplify this effect.

Finally, our data demonstrates that with a proper selection of parameter values, the hybrid model can achieve precision and recall performance virtually identical to the HMM Bayesian

approach alone, while providing those results in a fraction of the time. Similarly, the pruning operation employed in the HMMs has no noticeable positive or negative effect on the precision recall curves while providing a considerable improvement in runtime performance.

Regarding proper choice of parms, we note that most of these error rates are generally stable experimentally, as they arise from well-behaved physicochemical behaviors, such as photo-bleaching and chemical destruction of dyes (both easily measured from control experiments) and Edman rates (again measurable from control experiments). Several rates are inferred from observation (e.g. peptide detachment rates) are less easy to isolate in control experiments. However, we have generally observed high stability in these rates unless we are specifically manipulating them in control experiments (e.g. omitting key reagents to test Edman rates, etc).

A number of analytical techniques common in related fields were not explored. In tandem mass spectrometry (MS/MS), peptide spectral mapping is typically done either through data-base lookups and/or the use of simulated outcomes. Simulated mass spectra, consisting of ion pairs for the C- and N- terminal fragments for each potential breakage point, can be compared to the real data collected from the instrument [13]. Recent advances have been achieved by using deep learning to predict fragmentation behavior with higher quality than is possible with more traditional methods [16,17]. While we use the notion of matching fluorosequencing reads to a reference database, the specific algorithms are distinct.

Nevertheless, of possible relevance from the field of MS/MS is the analysis of the false discovery rate (FDR) [18,19]. The FDR is affected by two distinct sources: a peptide may be misattributed to the wrong peptide, even when the true peptide is present in the reference dataset, and MS/MS datasets contain significant amounts of modified peptides or contaminants, whose spectra may be mistakenly assigned to peptides in the reference set [20]. FDR is typically evaluated using a decoy database, such as is generated using reversed proteins from the target database. The FDR can then be set by referring to the number of hits in the decoy database given a particular score, as the decoy database is designed such that it should in theory never have hits for the biological sample being analyzed [21]. While an estimate of FDR based in theoretical analysis of the problem could find the misattribution rate of true peptides, even this estimate would be incomplete, because there are errors in mass spectra of peptides that cannot be accounted for by existing theory; furthermore, any effect of modifications or contaminants would likely be omitted.

The utility of a similar decoy database strategy for estimating FDR for fluorosequencing is unknown and remains to be established. We note however that, due to the rigorous probabilistic nature of our analysis, a reasonable estimate of FDR can be performed by subtracting the sum of PRM scores from the number of PRMs. This is the same as one minus the precision in a predicted precision/recall curve, and the proximity of our predicted precision/recall curve to the real curve for a known dataset demonstrates the feasibility of this approach (Fig 11). This analysis likely fails to account for the contributions of modifications and contaminants. We therefore plan to explore this problem more extensively in future work.

We also considered techniques for DNA sequence reconstruction. In general, DNA sequencing provides *de novo* sequence reconstructions and does not use reference database matching, and therefore is not a good model for fluorosequencing. Nevertheless, base calling strategies may have some relevance. For example, methods for base-calling from conventional (e.g. Illumina style) DNA sequencing are straightforward [22,23], and although errors occur, they are rare [24]. Analysis of errors in DNA sequencing is typically performed using multiple sequence alignment or k-mer based methods [25]. Because the error rates are typically much lower in DNA sequencing than in fluorosequencing, we believe existing software is unlikely to be effective in this new domain.

Nanopore DNA sequence analysis methods could also be considered. Nanopores, similar to fluorosequencing, deal with single molecule data and the concomitant statistical noise that process involves. However, nanopore data is on a real time continuum, with a DNA fragment which may move through the nanopore at variable rates during sequencing. Base-calling, the assignment of nucleic acid bases to chunks of sequencing information, is again the step most analogous to fluorosequencing. State of the art base-calling methods for nanopore sequencing typically use either HMMs or recurrent neural networks (RNNs). Comparisons of existing approaches suggest that RNNs slightly outperform HMMs in this domain [26]. While this suggests that RNNs are worth exploring for fluorosequencing data, we have avoided this approach for two reasons. First, RNNs are a deep learning technique, which invariably requires access to massive amounts of data; this is not currently feasible with fluorosequencing unless that data is simulated. Second, our approach suggests the possibility of direct estimation of parameters using some variant of the Baum-Welch algorithm adapted to our use case, which we believe would be significantly more difficult in an RNN based approach [27].

Tandem mass spectrometry and other types of proteomics datasets may require splitting training and testing sets into sets of non-overlapping peptides or proteins. This approach is critical when handling real data, because discrepancies between real and theoretical models are inevitable, and having different peptides in the training and testing data ensures that the resulting statistical corrections are capturing general properties of all peptides rather than overfitting the training data. Although this will be necessary to do for this model to correctly handle real data this is outside the scope of this paper.

## Conclusions

We have developed a powerful computational tool for the analysis of protein fluorosequencing data, which significantly increases the complexity of applications available to this new technology. This tool includes critically important optimizations which make our approach feasible in practice.

In future work we plan to implement a variation of the Baum-Welch algorithm to fit the parameters to data from a known peptide. We also wish to explore peptide and protein inference methods using peptide data classified using these methods. We may also explore *de novo* recognition of labels without use of a reference database.

## Supporting information

**S1 Table. Reference table for variables used throughout the supplemental appendices (S1-S4 Appendices).**
(PDF)

**S1 Appendix. HMM state space reduction.**
(PDF)

**S2 Appendix. Transition matrix factoring.**
(PDF)

**S3 Appendix. HMM pruning.**
(PDF)

**S4 Appendix. Combining transition matrix factoring with HMM pruning.**
(PDF)

**S1 Fig. Tuning the pruning parameter of the Bayesian HMM classifier.** Setting this parameter to 5 (i. e., 5 $\sigma$) appears to provide the best trade-off. (A) Precision/recall curves. We note

that "prune = 5" and "prune = 10" are hidden under the "prune = inf" curve. The "prune = 1" curve contains no true positives. (B) Runtimes.
(TIF)

**S2 Fig. Tuning the pruning parameter of the Bayesian HMM classifier.** Here we show a more extraordinary case than S1 Fig. (A) Precision/recall curves. We note that "prune = 5" and "prune = 10" are hidden under the "prune = inf" curve. The "prune = 1" curve contains no true positives. (B) Runtimes.
(TIF)

**S3 Fig. Tuning the k parameter of the kNN classifier.** Setting $k$ to 10 seems to provide the best trade-off (A) Precision/recall curves. (B) Runtimes.
(TIF)

**S4 Fig. Tuning the $\sigma_{kNN}$ parameter of the kNN classifier.** Setting $\sigma_{kNN}$ to 0.5 seems to provide the best trade-off. All settings showed a classify time of about 35 seconds, and a setup time of about 140 seconds.
(TIF)

**S5 Fig. Tuning the pruning parameter of the hybrid classifier.** A value of 5 appeared to provide the best trade-off. (A) Precision/recall curves. We note that "prune = 5" and "prune = 10" are hidden under the "prune = inf" curve. The "prune = 1" curve contains no true positives. (B) Runtimes.
(TIF)

**S6 Fig. Tuning h for the hybrid classifier.** An h of 1000 provided the best trade-off. (A) Precision/recall curves. (B) Runtimes.
(TIF)

**S7 Fig. Tuning k for the hybrid classifier.** A k of 1000 or 10000 provided the best results. (A) Precision/recall curves. (B) Runtimes.
(TIF)

**S8 Fig. Tuning the $\sigma$ parameter of the hybrid classifier.** Setting $\sigma$ to 0.5 seemed to provide the best trade-off. All settings had a classify time of about 50 seconds, and a setup time of about 140 seconds.
(TIF)

**S9 Fig. Sensitivity analysis of Edman failure rate.** The model expects an Edman failure rate of 6%, the rate for the test data varies as in the legend. Run on a full human proteome dataset.
(TIF)

**S10 Fig. Sensitivity analysis of detachment rate.** The model expects a detachment rate of 5%, the rate for the test data varies as in the legend. Run on a full human proteome dataset.
(TIF)

**S11 Fig. Sensitivity analysis of dud-dye rate.** The model expects a dud-dye rate of 7%, the rate for the test data varies as in the legend. Run on a full human proteome dataset.
(TIF)

**S12 Fig. Sensitivity analysis of bleach rate.** The model expects a bleach rate of 5%, the rate for the test data varies as in the legend. Run on a full human proteome dataset.
(TIF)

**S13 Fig. Sensitivity analysis of mu.** The model expects a mu value of 1.0, the rate for the test data varies as in the legend. Run on a full human proteome dataset.
(TIF)

**S14 Fig. Sensitivity analysis of sigma.** The model expects a sigma value of .16, the rate for the test data varies as in the legend. Run on a full human proteome dataset.
(TIF)

**S15 Fig. Sensitivity analysis of background sigma.** The model expects a background sigma value of .00667, the rate for the test data varies as in the legend. Run on a full human proteome dataset.
(TIF)

## Acknowledgments

The authors gratefully acknowledge Jagannath Swaminathan, Angela Bardo, Brendan Floyd, Daniel Weaver, and Eric Anslyn for helpful guidance and discussion throughout the course of this project. The authors acknowledge the Texas Advanced Computing Center (TACC) at The University of Texas at Austin for providing HPC (high performance computing) resources that have contributed to the research results reported within this paper. URL: http://www.tacc.utexas.edu. The authors additionally acknowledge usage of the jemalloc memory allocator which was crucial to achieving high performance results at scale (https://jemalloc.net/). We also acknowledge the Boost C++ test library (https://www.boost.org/doc/libs/1_82_0/libs/test/doc/html/index.html), and the FakeIt library by Eran Pe'er (https://github.com/eranpeer/FakeIt), both of which were used for unit testing our code. We additionally acknowledge use of the cxx-opts library by jarro2783 for parsing command line options (https://github.com/jarro2783/cxxopts), and use of Lohmann, N.'s json parsing library for handling parameter files (Lohmann, N. (2022). JSON for Modern C++ (Version 3.11.2) [Computer software]. https://github.com/nlohmann/json)

## Author Contributions

**Conceptualization:** Matthew Beauregard Smith, Zack Booth Simpson, Edward M. Marcotte.

**Formal analysis:** Matthew Beauregard Smith.

**Funding acquisition:** Edward M. Marcotte.

**Investigation:** Matthew Beauregard Smith, Zack Booth Simpson.

**Methodology:** Matthew Beauregard Smith.

**Resources:** Edward M. Marcotte.

**Software:** Matthew Beauregard Smith, Zack Booth Simpson.

**Supervision:** Edward M. Marcotte.

**Visualization:** Matthew Beauregard Smith.

**Writing – original draft:** Matthew Beauregard Smith.

**Writing – review & editing:** Edward M. Marcotte.

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
