## [Decision Letter · Decision Letter 0]

20 Mar 2023

Dear Smith,

Thank you very much for submitting your manuscript "Amino acid sequence assignment from single molecule peptide sequencing data using a two-stage classifier" for consideration at PLOS Computational Biology. As with all papers reviewed by the journal, your manuscript was reviewed by members of the editorial board and by several independent reviewers. The reviewers appreciated the attention to an important topic. Based on the reviews, we are likely to accept this manuscript for publication, providing that you modify the manuscript according to the review recommendations.

The reviewers are split in their opinion on your manuscript. Reviewer 1&2 are both very favorable, while reviewer 3 expresses concerns about the data you use to validate your model. While inclusion of experimental validation is not required for publication in PLOS Computational Biology, it should be referenced where possible. If reviewer 3 is right, it should be possible in this case. Please address the reviewers concerns in an updated manuscript.

Sincerely,

Lukas Käll

Guest Editor

PLOS Computational Biology

Ilya Ioshikhes

Section Editor

PLOS Computational Biology

The reviewers are split in their opinion on your manuscript. Reviewer 1&2 are both very favorable, while reviewer 3 expresses concerns about the data you use to validate your model. While inclusion of experimental validation is not required for publication in PLoS Computational Biology, it should be referenced where possible. If reviewer 3 is right, it should be possible in this case. Please address the reviewers concerns in an updated manuscript.

Reviewer's Responses to Questions

**Comments to the Authors:**

Reviewer #1: The paper proposes a two-stage peptide classification algorithm for data from florosequencing. The two stages consist of a k-nearest neighbor (kNN) classifier that can select a moderate number of candidate sequences whose data likelihood can be (approximately) computed by the second step using a hidden Markov Model for the data and turned into approximate posterior sequence probabilities by Bayes rule and summing of the kNN candidate sequences.

The paper is excellent. The idea is good and well-explained, the results are impressive, and the writing is impeccable. The paper can be accepted in its current form, and I have embarrassingly little feedback to give. A few short notes are provided below in the order in which the points appear in the paper, but I would like to leave it up to the authors' discretion what (if any) changes to make due to these, as most of them are mainly stylistic.

Line 104: "0.16" -> ".16" to match the prior style?

Line 120: "Each fluorophore count is stored as a two-byte numeric value." I found this note overly specific and think that it can be removed. It is an implementation detail in a section that is about concepts. The same comment applies to line 124 about double-precision floating-point values.

Line 138: Why you use the word "approximated" here is unclear. I understand the approximations introduced later, but here you are describing the exact transition probabilities the model gives. Do you mean that the model itself is an approximation? Consider reformulating this statement.

Line 150: To be a bit pedantic, (2) should not be a *probability* given that Y is a continuous random variable. Maybe this can be dealt with by simply adding a footnote at (2) and then leaving the rest of the text unmodified.

Line 172: It was initially unclear to me what you considered to be the default states, from which you later obtained the reduced (more inclusive) states. Can you clarify this? If so, please do. Note that you, on line 646, then write "the true state," which suffers the same problem. Why would this state be more "true" than the reduced set of states?

Line 194: Why do you write "improve the theoretical complexity"? The practical complexity is also reduced by the introduced means. If you mean that the complexity order (as in the Big-O notation) is reduced, then it is better to state this as "complexity order" or similar.

Lines 252 and 253: The "two-byte" and "eight-byte" comments are, like before, too detailed for this text. They are implementation details that can be excluded from this conceptual text.

Line 258: "the k nearest dye track neighbors." You need to specify the norm used to define nearest here. Even if it is just the standard Euclidean norm, please say so.

Line 315: This line refers to the "Monte Carlo simulation section of the paper," but there are no sections named this.

Line 320: I do not think the notion of "a cut-off value" is defined in this context. It becomes clear later by reading the appendices, but you should ensure the main text is understandable without the appendices or explicitly referring to them for the definition.

Line 679: "the matrix vector" -> "The matrix-vector."

Line 708: "color of fluorophore" -> "color of the fluorophore"

Line 786: You should specify that you consider the evaluation of (31) for all t to get the T in the Big-O expressions.

Line 787: When writing O(r^2T) you should clarify that you disregard the complexity of sorting the elements (which would be on the order of SlogS where S is the number of states). I understand that your later strategy does not need to sort any entries explicitly, but this needs to be clarified at this text stage.

Line 853: Consider writing "\\sigma_{kNN}" instead of "kNN \\sigma" in Figure B4 to be consistent with prior notation.

Reviewer #2: Single molecule protein sequencing is a potentially revolutionary technology that could disrupt life sciences as we know it. Current methods for protein sequencing such as those based on Edman degradation rely on large ensemble measurement, whereas the authors have developed in a previous study a method whereby single molecule fluorescence detection of immobilized peptides undergoing cyclic Edman degradation could be used to identify proteins in a massively parallel format analogous to what is done with DNA on an Illumina chip. In this paper, the authors have developed a computational followup that addresses the issue of deconvoluting the noisy and complex data that arises from these fluorosequencing experiments – a combination of multiple fluorescently labeled amino acids on a randomly cleaved peptide fragment undergoing a series of Edman degradation cycles. The authors used a hidden Markov model (HMM) to model the underlying process of degradation, fluorescence photobleaching, and random peptide detachment along with the observable – a fluorescence intensity signal at each cycle. Because the traditional HMM approach of assigning outcomes via Bayes theorem scales poorly with the vast number of possible peptide sequences and possible states, the authors developed a hybrid approach that utilizes the computationally efficient kNN classifier to reduce the complexity of the problem and apply Bayesian classification on a limited subset of possible sequences. The context of the paper, the fluorosequencing technology, is exciting and addresses a major gap in biotechnology. This paper adds a novel contribution beyond the contents of their previous work. It is also well written and technically sound. I have some minor points which I think could either be discussed and clarified in text, or if judged appropriate, be addressed experimentally to improve the manuscript.

You’ve said that you will use the Baum Welch algorithm in the future to estimate HMM parameters – but in the meantime your model requires prior knowledge of those parameters. How much variation do you expect in the parameter values (state transition probabilities etc) and how does this propagate into classification performance? While describing your Bayesian classifier on line 137 you wrote “Transition probabilities between these states can be approximated using previously estimated success and failure rates of each step of protein fluorosequencing.” Is this realistic, and if these values are assigned with 5, 10, 50% error – how will this affect the precision/recall relations downstream?

Along the above lines – do you know if these parameters are stable? Is it realistic that the transition probabilities and other parameters are going to be fixed and reproducible from experiment to experiment or even as time evolves within an experiment? Wondering how brittle or robust the peptide-calling is, and how much depends on accurate parameterization which, I can imagine, might be resource intensive if it must be done with every experiment.

On line 418 you wrote: “ It is also interesting that the HMM pruning operation is more necessary with longer peptides and more colors of fluorophores; with the trypsinized dataset labeling D/E, C, and Y, omitting pruning had little consequence, but in moving to cyanogen bromide with D/E, C, Y, and K, we observed a runtime speedup of about 1000-fold.” Could you explain the intuition behind this scaling phenomenon?

Reviewer #3: The paper describes a model for peptide sequence determination from the data produced from fluorosequencing - a new recently proposed technology. The authors constructed several different machine learning models, using combinations of HMM and kNN classifiers. The fluorosequencing data was simulated based on the model proposed in the previous manuscript by the same group. The authors trained and evaluated multiple prediction models using the simulated data and concluded that a hybrid HMM/kNN model achieved the balance between precision and performance. The manuscript is well-written, the computational problem is clearly explained, and the proposed models are sound. The model implementations are publicly available on github.

My only concern is the lack of testing on the real fluorosequencing data, given that the overlapping group of authors have generated such data in their previous work. Since the current models were trained and evaluated under the same Monte-Carlo simulation model, it is unclear how the proposed models would extend to the real measurements. In addition, I was not able to find how the authors split their simulations into training/testing sets. Did training and testing sets contain non-overlaping peptides?

**Have the authors made all data and (if applicable) computational code underlying the findings in their manuscript fully available?**

Reviewer #1: Yes

Reviewer #2: Yes

Reviewer #3: Yes

PLOS authors have the option to publish the peer review history of their article (what does this mean?). If published, this will include your full peer review and any attached files.

Reviewer #1: No

Reviewer #2: **Yes: **Ian T Hoffecker

Reviewer #3: No

Figure Files:

Data Requirements:

Reproducibility:

References:

---

## [Editor Report · Decision Letter 1]

4 May 2023

Dear Matthew Beauregard Smith,

We are pleased to inform you that your manuscript 'Amino acid sequence assignment from single molecule peptide sequencing data using a two-stage classifier' has been provisionally accepted for publication in PLOS Computational Biology.

We see that you have responded to the reviewers comments in a satisfactory way, and I do not see any reasons to delay the manuscript's acceptance. However, we noted that you introduce the word 'log' in line 149 in the revised manuscript, which can be interpreted in multiple ways in this context. It could mean log as keeping track of, and log as calculating the logarithm. Please consider replacing the word in the proofs.

Best regards,

Lukas Käll

Guest Editor

PLOS Computational Biology

Ilya Ioshikhes

Section Editor

PLOS Computational Biology

---

## [Editor Report · Acceptance letter]

24 May 2023

PCOMPBIOL-D-23-00097R1 

Amino acid sequence assignment from single molecule peptide sequencing data using a two-stage classifier

Dear Dr Smith,

I am pleased to inform you that your manuscript has been formally accepted for publication in PLOS Computational Biology. Your manuscript is now with our production department and you will be notified of the publication date in due course.

With kind regards,

Judit Kozma
